# A Dataset To Evaluate The Representations Learned By Video Prediction Models

**Ryan Szeto**[1,2,*]   **Simon Stent**[1]   **German Ros**[1]   **Jason J. Corso**[2]
[1] Toyota Research Institute, Cambridge, MA    [2] University of Michigan, Ann Arbor, MI
`{szetor,jjcorso}@umich.edu`
`{simon.stent,german.ros}@tri.global`

## Abstract

We present a parameterized synthetic dataset called **Moving Symbols** to support the objective study of video prediction networks. Using several instantiations of the dataset in which variation is explicitly controlled, we highlight issues in an existing state-of-the-art approach and propose the use of a performance metric with greater semantic meaning to improve experimental interpretability. Our dataset provides canonical test cases that will help the community better understand, and eventually improve, the representations learned by such networks in the future. Code is available at `https://github.com/rszeto/moving-symbols`.

## 1 Introduction

The question of whether modern video prediction models can correctly represent important sources of variability within a video, such as the motion parameters of cameras or objects, has not been addressed in sufficient detail. Moreover, the manner in which existing video prediction datasets have been constructed—both real-world and synthetic—makes probing this question challenging. Real-world datasets have thus far used videos from action recognition datasets (Soomro et al., 2012; Karpathy et al., 2014) or other "in-the-wild" videos (Santana & Hotz, 2016), assuring to some degree that the sets of sampled objects and motions will vary realistically between training and testing time. However, as the parameters governing the video appearance and dynamics are unknown, failures in prediction cannot be easily diagnosed. Synthetic datasets, on the other hand, define a set of dynamics (such as object translation or rotation) and apply them to simple objects with pre-defined appearances such as MNIST digits (LeCun et al., 1998) or human head models (Singular Inversions, 2017). Since they are generated from known sets of objects and motion parameters, representations from a video prediction model can be evaluated based on how accurately they predict the generating parameters. However, up until now, both the training and testing splits of these datasets have been generated by sampling from the same object dataset and set of motion parameters, making it difficult to gauge the robustness of video representations to unseen combinations of object and dynamics.

Our proposition is simple: a dataset that explicitly controls for appearance and motion parameters—and can alter them between training and testing time—is essential to answering the question of whether video prediction networks capture in their representations useful physical models of the visual world. To this end, we propose the Moving Symbols dataset, which extends Moving MNIST (Srivastava et al., 2015) with features designed to answer this question. Unlike existing implementations of Moving MNIST, which hard-code the image source files and speed of dynamics, we allow these parameters to change between training and testing time, enabling us to evaluate a model's robustness to multiple types of variability in the input. Additionally, we log the motion parameters at each time step to enable the evaluation of motion parameter recovery.

To demonstrate the utility of our dataset, we evaluate a state-of-the-art video prediction model from Villegas et al. (2017) on a series of dataset splits designed to stress-test its capabilities. Specifically, we present it with videos whose objects exhibit appearances and movements that differ from the training set in a controlled manner. Our results suggest that modern video prediction networks may fail to maintain the appearance and motion of the object for unseen appearances and rates, a problem that, to the best of our knowledge, has not yet been clearly expounded.

---

[*]This work was completed while Ryan Szeto was an intern at Toyota Research Institute.

| Experiment group | Train images | Test images | Training rates | Testing rates |
|---|---|---|---|---|
| (1a) Translation speed variation, slow → fast | MNIST | MNIST | ▶ | ▶▶ |
| (1b) Translation speed variation, fast → slow | MNIST | MNIST | ▶▶ | ▶ |
| (1c) Translation speed variation, slow/fast → med | MNIST | MNIST | ▶ ▶▶ | ▶▶ |
| (2a) Appearance variation, MNIST → Icons8 | MNIST | Icons8 | ▶ ▶▶ ▶▶ | ▶ ▶▶ ▶▶ |
| (2b) Appearance variation, Icons8 → MNIST | Icons8 | MNIST | ▶ ▶▶ ▶▶ | ▶ ▶▶ ▶▶ |

▶ Slow    ▶▶ Medium    ▶▶ Fast

Table 1: Experimental setup. The row for each trial describes the objects and motion speeds seen during training and the out-of-domain objects and motion speeds seen during testing.

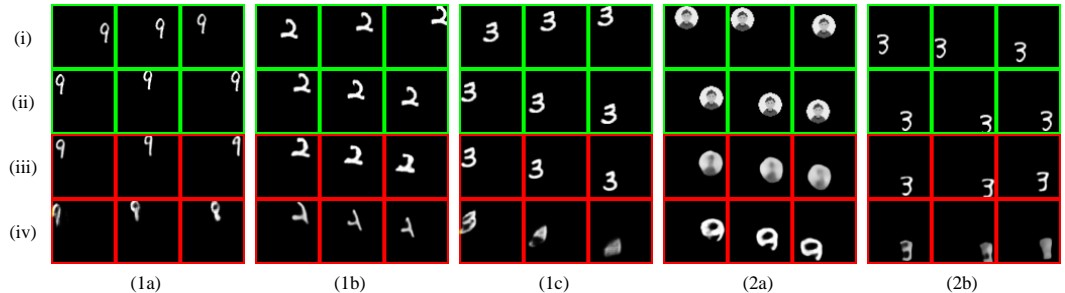

(1a)          (1b)          (1c)          (2a)          (2b)

Figure 1: Sample predictions for each experiment from Table 1. From top to bottom: (i) sample input frames observed by each prediction model at $t = \{3, 6, 9\}$; (ii) sample ground truth frames (unobserved) at $t = \{13, 16, 19\}$; (iii) corresponding predictions from a model tested under the same conditions as the training set; and (iv) predictions from a model tested under different conditions, as per Table 1.

## 2 MOVING SYMBOLS

The Moving MNIST dataset (Srivastava et al., 2015) is a popular toy dataset in the video prediction literature (Patraucean et al., 2015; Kalchbrenner et al., 2017; Denton & Birodkar, 2017). Each video in the dataset consists of one or two single digit images that translate within a $64 \times 64$ pixel frame with random velocities. Although recent approaches can generate convincing future frames, they have been both trained and evaluated on videos generated by the same object dataset and motion parameters, limiting our understanding of what the networks learn to represent.

Moving Symbols overcomes this limitation by allowing sampling from *different* object datasets or sets of motion parameters at training and testing time. For example, the training set may contain slow- and fast-moving MNIST digits while the test set may contain Omniglot characters (Lake et al., 2015) at medium speed. By adjusting a configuration file, it is trivial to generate paired train/test splits under varying conditions. Furthermore, the video generator logs the image appearance, pose, and rate of movement of each object at each time step, which enables semantically meaningful evaluation as we demonstrate in Section 3.

## 3 EXPERIMENTS

To demonstrate the insights we can gain from the Moving Symbols dataset, we construct two groups of experiments, where each trial consists of different train/test splits (see Table 1). The experiments are designed to elucidate the model's ability to generalize across variation in either motion (1a-c) or appearance (2a-b). For appearance, to demonstrate the flexibility of our dataset generator, we introduce a novel dataset of 5,000 vector-based icons from icons8.com. We use our evaluation framework to analyze MCNet (Villegas et al., 2017), a state-of-the-art video prediction model. For each experiment, we generate 10,000 training videos and 1,000 testing videos. We train the model on each unique training set, using ten frames of input to predict ten future frames. After training the model for 80 epochs using the same procedure as Villegas et al. (2017), we evaluate it on the out-of-domain test set with ten input frames and twenty predicted future frames.

Figure 2: Quantitative results to compare in-domain (dotted line) and out-of-domain evaluation performance (solid line), with standard error shown in gray. Across future time steps, we report median values for two metrics: positional Mean Squared Error (MSE) between the oracle's predicted position and ground truth (first and third plots), and cross-entropy between the oracle's digit prediction and true label (second and fourth plots). Lower is better in all cases.

Figure 1 shows a comparison of the predictions made for a random test sequence in each experiment. For the speed variation experiments, we observe that when MCNet is evaluated on out-of-domain videos, it propagates motion by replicating trajectories seen during training rather than adapting to the speed seen at test time (e.g. in experiment (1a) from Table 1, MCNet slows down test digits). More surprisingly, the model has difficulty preserving the appearance of the digit when motion parameters change. Since MCNet explicitly takes in differences between subsequent frames as input, it is possible that the portion of the model encoding frame differences inadvertently captures and misrepresents appearance information. In the appearance variation experiments, MCNet clearly overfits to the appearances of objects seen during training. For example, the MNIST-trained model transforms objects into digit-like images, and the Icons8-trained model, which sees a broader variety of object appearances, transforms digits into iconographic images. This may be due to the use of an adversarial loss to train MCNet, which would penalize appearances that are not seen during training.

To obtain quantitative results, we train an "oracle" convolutional neural network to estimate the digit class and location of MNIST digits in a $64 \times 64$ pixel frame and compare them against the logged ground truth. This allows us to draw more interpretable comparisons between generated frames than is possible with common low-level image similarity metrics such as PSNR or SSIM (Wang et al., 2004). For the oracle's architecture, we use LeNet (LeCun et al., 1998) and attach two fully-connected layers to the output of the final convolutional layer to regress the digit's location. On a held-out set of test digits, the trained oracle obtains 98.47% class prediction accuracy and a Root Mean Squared Error (RMSE) of 3.5 pixels for 64x64 frames.

Figure 2 shows our quantitative results on the MNIST-based test sets (all rows except (2a) from Table 1). For the speed variation experiments, the frames predicted for out-of-domain test videos induce a large MSE from the oracle, especially during later time steps, whereas in-domain evaluation yields frames that induce a small MSE. This matches our qualitative observation that MCNet fails to propagate the out-of-domain object's motion, instead adjusting it to match the speeds seen during training. The digit prediction performance of the oracle also drops substantially when evaluating on out-of-domain videos, supporting our observation that MCNet has trouble preserving the appearances of objects that move faster or slower than those seen in training. In the appearance variation experiment (2b), we observe a large digit classification cross-entropy error for the out-of-domain case compared to the in-domain case because the Icons8-trained model transforms digits into iconographic objects. Surprisingly, the model preserves the trajectory of unseen digits better than the MNIST-trained model. This might be because predicting the wrong location for pixel-dense Icons8 images would incur a larger pixel-wise penalty during training than predicting the wrong location for pixel-sparse MNIST images.

## 4 DISCUSSION AND FUTURE WORK

We have shown that Moving Symbols can help expose the poor generalization behavior of a state-of-the-art video prediction model, which has implications in the real-world case where unseen objects and motions abound. Only a fraction of our dataset's functionality has been demonstrated here—its other features can be used to construct more elaborate experiments, such as introducing scale, rotation, and multiple symbols with increasing complexity, or making use of the pose information as a further supervisory signal. The community's adoption of this dataset as an objective, open standard would lead to a new generation of self-supervised video prediction models. The Moving Symbols dataset is currently available at `https://github.com/rszeto/moving-symbols`.

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
