# OpenReview forum: "A Dataset To Evaluate The Representations Learned By Video Prediction Models"
_ICLR.cc/2018/Workshop — Accept_

### Official Review · AnonReviewer2 · 2018-03-07
**A  means to allow for systematic evaluation of video-based methods**

**Rating:** 9
**Confidence:** 4

**Review:**


==========================================
Summary
==========================================

The paper proposes Moving Symbols, a means to generate synthetic video datasets with the capability of controlling changes on apperance and motion of objects occurring in the video. The main goal of the proposed dataset is to allow systematic experimentation of video prediction methods.

----------------
Novelty
----------------

Even though the proposed dataset is an extension of the Moving MNIST dataset (Srivastava et al., ICML'15), its contribution increases the benchmarking capabilities of the dataset.

----------------
Clarity
----------------

The paper is clear and its content is easy to follow.

----------------
Significance
----------------

The proposed dataset is a timely addition to recent interest on video prediction.
In addition, it allows for the systematic evaluation of these methods.

----------------
Quality
----------------

The manuscript displays good quality. Contributions are properly motivated, the evaluation protocol is clear and observations made on the evaluation are adequately discussed.


==========================================
PROs:
- Clear and easy to follow.
- Need of the proposed dataset is properly motivated.
- Observations of the conducted experiments are adequately discussed.
- Evaluation code is made available.


CONS:
- Even though the evaluation is appropriate, it is limited to a single existing method, i.e. MCNet (Villegas et al. ICLR'17).
==========================================

---

> ### Author Response · Authors · 2018-03-29
> **Authors' Response to AnonReviewer2**
>
> Thank you for your comments. Addressing your concern with limiting our analysis to a single existing method, we focused our analysis on the best-performing video prediction method whose code is publicly available (to the best of our knowledge) due to space constraints. However, we agree that comparing several state-of-the-art models would strengthen our conclusions. We welcome authors of new video prediction models to evaluate and objectively compare the performance of their models on versions of our dataset with published parameters.

---

### Official Review · AnonReviewer3 · 2018-03-11
**a dataset with controllable amount of train/test motion-content differences for video prediction**

**Rating:** 7
**Confidence:** 5

**Review:**

A dataset is introduced where appearance and motion statistics are varied in a controllable manner between train and test time, and various splits are proposed to test capabilities of current video prediction methods to generalize to novel motion or appearance profiles between train and test.
The dataset is tested on the model of Villegas et al 2017.
The conclusion is it does not generalize to novel content or motion profiles.
Though controlling such train/test dissimilarity statistics seems like a great idea, it is not clear that a learning algorithm or even the human brain can or should generalize to much novel motion profiles or appearances. This though is a different discussion. i find the dataset very valuable for evaluation purposes.

---

> ### Author Response · Authors · 2018-03-29
> **Authors' Response to AnonReviewer3**
>
> Thank you for your comments; we are happy that you find our dataset very valuable. Addressing your comment about justifying generalizability --- specifically that "it is not clear that a learning algorithm or even the human brain can or should generalize to much novel motion profiles or appearances" --- we agree but, as mentioned in the introduction, real-video datasets used for video prediction do exhibit (to varying extent) novel motion profiles or appearances between training and testing (e.g. training on a video of a professional baseball pitcher and testing on a video of an amateur pitcher in casual clothing). As our dataset allows researchers to control this domain gap, it is a step towards better quantification and understanding of the generalizability problem. It also allows us as a community to ask how we should modify existing learning algorithms to more efficiently adapt to unseen domains that they struggle to immediately generalize to.

---

### Official Review · AnonReviewer1 · 2018-03-18

**Rating:** 7
**Confidence:** 4

**Review:**

This paper proposes a new dataset, which can be seen as a variation (in spirit at least) of the Moving MNIST dataset (Srivastava et al., 2015). Specifically, instead of having the same data classes in both training and test set, the proposed dataset has different ones (e.g., MNIST like in training, Omniglot like in test). The reason is that by having the same "semantic" distributions in both training and test sets, it's harder to judge whether the semantics themselves influence the learning of dynamics of objects over time. What is more, different from Moving MNIST, where the  image source files and speed of dynamics are hard coded, in this  paper they are sampled on the fly.

The dataset is an interesting contribution, and there is not much to say to make it better. I would note that sampling on the fly is better to evaluate the generalization capabilities of the algorithms. However, at the same time it makes comparisons between different methods from different papers somewhat arbitrary. It would be good if the authors offered also "comparison-friendly" dataset hyperparameters, so that new papers can compare with previous methods on equal grounds. Another variation that could also be considered is the scale of symbols, e.g., their size can vary over time. To make this more general, other geometric transformations can be considered, such as skewing, or even symbol morphings? Of course, in the end each variation should serve a purpose, namely what type of dynamics are tested here?

---

> ### Author Response · Authors · 2018-03-29
> **Authors' Response to AnonReviewer1**
>
> Thank you for your comments. We appreciate that you have considered our method to be better suited for evaluating the generalization capabilities of video prediction algorithms than prior work. Please find our responses to your comments below.
>
> 1. Regarding the "comparison-friendly" dataset hyperparameters, we note that the parameters used to generate the data for our experiments are included in our GitHub code. If the dataset is adopted by other researchers, we encourage them to submit pull requests for their own parameter configurations so that other authors can make direct comparisons.
>
> 2. With regard to varying the scale of symbols over time, this is a feature supported in our code; however, we omitted our experiments on this behavior due to space constraints.
>
> 3. "To make this even more general, other geometric transformations can be considered…": We agree that supporting other geometric transformations would further improve our understanding; skewing and distorted symbols can be incorporated in future work. As the goal of our dataset is to evaluate the extent to which video prediction models capture the underlying content- and motion-defining parameters that are shared between a training set and a held-out set, adding any visual transformation that exists within that other dataset would be justified.

---

### Decision · Program_Chairs · 2018-03-20
**ICLR 2018 Workshop Acceptance Decision**

**Decision:**

Accept

**Comment:**

Congratulations, your paper was accepted to the ICLR workshop.